# Taxifolin as a Metallo-β-Lactamase Inhibitor in Combination with Augmentin against Verona Imipenemase 2 Expressing *Pseudomonas aeruginosa*

**DOI:** 10.3390/microorganisms11112653

**Published:** 2023-10-28

**Authors:** Bogdan M. Benin, Trae Hillyer, Aylin S. Crugnale, Andrew Fulk, Caitlyn A. Thomas, Michael W. Crowder, Matthew A. Smith, Woo Shik Shin

**Affiliations:** 1Department of Pharmaceutical Sciences, Northeast Ohio Medical University, Rootstown, OH 44272, USA; bbenin@neomed.edu (B.M.B.); thillyer@neomed.edu (T.H.); afulk@neomed.edu (A.F.); msmith13@neomed.edu (M.A.S.); 2Department of Chemistry and Biochemistry, Miami University, Oxford, OH 45056, USA; thomas60@miamioh.edu (C.A.T.); crowdemw@miamioh.edu (M.W.C.); 3Akron Children’s Hospital, Rebecca D. Considine Research Institute, Akron, OH 44302, USA

**Keywords:** metallo-β-lactamases, computer-aided drug design, natural products, flavonol, taxifolin, quercetin, VIM-2, *Pseudomonas aeruginosa*, combination therapy, ESKAPE pathogens

## Abstract

Among the various mechanisms that bacteria use to develop antibiotic resistance, the multiple expression of β-lactamases is particularly problematic, threatening public health and increasing patient mortality rates. Even if a combination therapy—in which a β-lactamase inhibitor is administered together with a β-lactam antibiotic—has proven effective against serine-β-lactamases, there are no currently approved metallo-β-lactamase inhibitors. Herein, we demonstrate that quercetin and its analogs are promising starting points for the further development of safe and effective metallo-β-lactamase inhibitors. Through a combined computational and in vitro approach, taxifolin was found to inhibit VIM-2 expressing *P. aeruginosa* cell proliferation at <4 μg/mL as part of a triple combination with amoxicillin and clavulanate. Furthermore, we tested this combination in mice with abrasive skin infections. Together, these results demonstrate that flavonol compounds, such as taxifolin, may be developed into effective metallo-β-lactamase inhibitors.

## 1. Introduction

Since the discovery of β-lactam antibiotics more than five decades ago, a huge decline in mortality rates has occurred. These antibiotics still serve as important primary-line treatments against common bacterial infections; however, resistance to β-lactam antibiotics, especially among the ESKAPE pathogens (*Enterococcus faecium*, *Staphylococcus aureus*, *Klebsiella pneumoniae*, *Acinetobacter baumannii*, *Pseudomonas aeruginosa*, *Enterobacter* spp.), threatens global public health while every year costing upwards of USD 4 billion due to community and hospital-acquired infections and causing over 23,000 deaths [1]. One of the major drivers of this resistance is the expression of β-lactamase enzymes that cleave β-lactam rings, rendering β-lactam antibiotics inactive. β-lactamases are classified into four classes: serine (classes A, C, and D; SBL) and metallo (class B) β-lactamases (MBLs). The primary difference between these classes is that SBLs utilize a catalytic serine residue, whereas metallo-β-lactamases utilize metal-activated water as a nucleophile for β-lactam hydrolysis [2]. Combining β-lactamase inhibitors (BLis) with β-lactam antibiotics is the only effective way to overcome β-lactam drug resistance while using the current antibiotic arsenal. Today, three classes, with six different drugs, of FDA-approved SBLis exist based on β-lactams (sulbactam, tazobactam, and clavulanate), 1,6-diazabicyclo octanes (avibactam, relebactam), and boron-based (vaborbactam) scaffolds. Still, and critically, none of these available inhibitors are effective against the most common MBLs, such as New Delhi MBL 1 (NDM-1) or Verona imipenemase 2 (VIM-2), which have emerged as major threats in recent years [3,4,5,6,7].

To inhibit MBLs, two strategies targeting the active site have been devised: (i) zinc sequestration/stripping and (ii) direct binding to zinc in the active site (i.e., competitive inhibition or ternary complex formation) [8]. Although Zn stripping agents or very strong chelators (e.g., N,N,N′N′-ethylenediaminetetraacetic acid (EDTA) or aspergillomarasmine A) have been demonstrated as effective MBL inhibitors [9,10,11,12,13,14,15], these agents are non-specific, and several reports have detailed the evolution of mutated MBLs exhibiting higher Zn affinities and greater antibiotic resistance promoted by Zn-limited conditions [7,14,16,17].

As a result of these limitations, more attention has been given to the research and development of inhibitors that interact directly with the zinc at the active site without sequestration. These promise, through the greater selectivity of enzymatic active sites, to be safer and more effective. Currently, two such inhibitors are in phase 1 and phase 3 clinical trials: QPX7728 and taniborbactam, respectively [18,19,20]. These are both based on boronic acids and have demonstrated broad-spectrum MBL and SBL inhibition. Still, these inhibitors are not equally active against all MBL classes, their mechanisms remain unclear and, although highly promising, they remain under investigation [17].

Typically, potential candidates must be able to (i) act as weak metal chelators, (ii) prevent the degradation of β-lactam antibiotics to restore their primary activity, and (iii) access the active site of enzymes. Flavonols, and quercetin more specifically, have been sporadically reported, with various studies suggesting that they may exhibit beneficial biological activities such as anti-inflammatory, anti-cancer, anti-viral properties, etc. [21,22,23,24,25,26]. Now, however, these have received renewed attention as they have been demonstrated to successfully satisfy these three criteria for MBL inhibition (metal chelation, active binding site access, β-lactam cleavage prevention) and have been shown via solution NMR to bind to the active site of NDM-1 [21].

The first of these criteria, metal chelation, is achieved by the presence of various donor atoms on adjacent carbons. Although two possible binding sites exist (catechol and a β-hydroxyketone-like group), a Zn-quercetin complex was recently synthesized and demonstrates that quercetin acts as a bidentate ligand via the β-hydroxyketone-like functionality [27]. Importantly, quercetin binds Zn far weaker than other chelators and other metalloenzymes (Kd ~ 10^−4^), supporting the notion that it will be effective without metal-stripping and without disrupting other enzymes [28].

Reports on the second criterion, inhibition of the enzymatic cleavage of β-lactams, have thus far been scarce, although the concept of metalloenzyme inhibition by flavonols has been discussed since the 1990s [29,30]. One of the initial studies that focused on the ability of quercetin to act as an MBL inhibitor was published by West et al., in which quercetin and galangin were demonstrated to inhibit the MBL (L1) expressed by *Stenotrophomonas maltophilia* [31]. However, the inhibitor–enzyme conjugate structure was only demonstrated theoretically, and, although promising, the authors did not demonstrate the inhibition of β-lactam hydrolysis in bacterial studies.

Only recently, in 2020, was the third criterion demonstrated to be fulfilled by Morellet et al. who solved, for the first time, a flavonol-MBL (NDM-1) structure through the use of solution nuclear magnetic resonance (NMR) spectroscopy [21]. The authors solved the structures for several NDM-1-flavonol conjugates with quercetin, myricetin, and morin. This not only demonstrated that these could act as competitive NDM-1 inhibitors, but it also indirectly validated the earlier, theoretical placement of galangin in the L1 active site. The authors also demonstrated that quercetin and its analogs, once bound to the NDM-1 active site, could effectively block the hydrolysis of imipenem. Their work joins others in demonstrating the capability of flavonols to act as effective MBLis. Other works have also supported this finding by treating clinical isolates with quercetin-containing combination therapies; however, these have generally not been focused on the ability of quercetin to inhibit MBLs, and some studies utilized non-resistant strains or non-β-lactam antibiotics [32,33,34,35,36,37,38,39,40,41,42]. Of these, only Eumkeb et al. directly mentioned quercetin as a potential MBLi, but without reference to any specific MBL. This lack of control over MBL-type has insofar represented one of the key gaps in our current understanding of flavonol inhibitors, and it is unknown how well quercetin will interact with other MBLs (e.g., VIM, IMP, etc.).

Given the promising demonstration of NDM-1 inhibition by ternary complex formation with quercetin, myricetin, and morin, we sought to determine if this activity was also present against VIM-2—another class B1 MBL with widespread prevalence and culpable for a large portion of carbapenem treatment failures in *Pseudomonas aeruginosa* (PA) infection cases [3,43,44,45]. We have, therefore, focused our current efforts on PA expressing VIM-2, as this pathogen is not only highly prevalent in hospital-acquired infections, especially in the wake of the COVID-19 pandemic, but it is also known to harbor, share, and express this MBL. Even newly approved treatments, which have not yet been widely adopted throughout the world, such as cefidericol, are only effective against some MBL-producing PA strains [46], necessitating the continued discovery and development of safe and effective inhibitors.

Herein, we demonstrate that some quercetin–analog compounds may be promising scaffolds for the further development of combination therapies aimed at successfully treating VIM-2-producing PA. Our combined computational, in vitro, and in vivo study of a series of quercetin–analog compounds demonstrates that taxifolin, specifically, provides promise even without further development or optimization when combined with amoxicillin and clavulanate in treating VIM-2 expressing PA.

## 2. Materials and Methods

Quercetin, its analogs, and nitrocefin were purchased from the Cayman Chemical company (Ann Arbor, MI, USA). Ten mM stock solutions were prepared with H_2_O or DMSO (Sigma Alrdich, St. Louis, MO, USA, ≥99.9%). All materials were used as received without further purification. Assays were performed with biological triplicates.

### 2.1. Docking Studies

Modeling and docking studies were carried out with the Maestro Schrodinger 2022-2 software package (Schrodinger, Inc.; New York, NY, USA). Computational studies were based on NMR and X-ray structures (PDB: 6TTA and 4C1D). Proteins were prepared through the standard workflow, which includes adding missing sidechains and hydrogens as well as energy minimization using the OPLS-2005 force field. Glide docking was utilized to generate docking scores and compare models [47,48]. For each compound, a minimum of 5 poses were included. A two-tailed unpaired *t*-test was utilized to determine statistical significance. The resulting average, standard deviation, and *p*-value were reported.

### 2.2. Biochemical Assays

Nitrocefin was used as the substrate, as it is a chromogenic cephalosporin. The activity of purified VIM-2 was measured spectrophotometrically (Cytation 3, BioTek, Winooski, VT, USA) in a potassium phosphate buffer (PBS, pH = 7.4, prepared according to a published recipe) [49]. Nitrocefin concentrations ranging from 8 µM to 120 µM were utilized. The formation of the hydrolyzed product is measured at 486 nm at intervals of 10 s for at least 30 min until a plateau in product formation is observed [50,51,52]. The Km and kcat values were determined by plotting initial velocity measurements followed by fitting with a non-linear regression Michaelis–Menten kinetic model in OriginLab 2022b (OriginLab Corp., Northhampton, MA, USA). 

The inhibitor constants (K_i_) were determined following a similar approach, where the inhibitor was fixed at 500 μM with nitrocefin ranging from 5 to 100 μM. A blank consisting of DMSO, PBS, and the inhibitor was utilized to subtract any potential absorbance from the colored inhibitors at 486 nm. Each inhibitor was pre-incubated with the VIM-2 enzyme for 10 min at room temperature, and then the absorbance at 486 nm was measured and evaluated using a competitive inhibition model in Origin 2022b using the previously determined Km value.

Single-dose enzyme inhibition was determined by comparing the absorbance of the hydrolyzed nitrocefin (λ = 486 nm) after several time points (e.g., 5 min, 30 min, 1 h) between various samples of purified TEM-1 (10 nM) with and without inhibitors. Quercetin analog inhibitors were tested at concentrations of 500 μM, whereas known covalent inhibitors (clavulanate, sulbactam, tazobactam, vaborbactam) had concentrations of 50 μM. Enzymes were pre-incubated with inhibitors for 10 min at room temperature prior to their addition to nitrocefin (final nitrocefin concentration of 120 μM).

### 2.3. Bacteria Transformation

Starting with bioluminescent PA (PAX5), we developed a VIM-2 expressing strain of PAX5 (PAX5_VIM-2_). VIM-2 plasmid was isolated using the protocol provided by New England BioLabs, Inc. with their Monarch Plasmid Miniprep Kit. The transformation was carried out following a previously published procedure utilizing electroporation [53]. After incubation in recovery media, bacteria (100 μL) were plated on Luria–Bertani Agar (LBA) containing 50 μg/mL of kanamycin and incubated overnight at 37 °C. The following day, samples were taken and added to LB broth and incubated overnight at 37 °C with agitation (120 rpm). To test whether VIM-2 was expressed, disc diffusion assays with ethylenediaminetetraacetic acid (EDTA) and single-dose inhibition assays were carried out.

### 2.4. Disc Diffusion Assay

Bacteria cultured overnight were diluted to an optical density of 0.08–0.1 at 600 nm. Kirby–Bauer discs containing either 10 μg meropenem or 10 μg meropenem with 930 μg EDTA were added to the inoculated Müller–Hinton Agar (MHA) plates and incubated overnight at 37 °C [54,55,56]. The following day, photos and luminescence images were recorded to determine the inhibition zones around each disc.

### 2.5. Single-Dose Inhibition Assay

Bacteria were diluted to 5 × 10^5^ CFU/mL in MH broth (MHB). A growth control containing only bacteria and a negative control containing only MHB were added. Test wells were treated with either 25 µM meropenem, ceftazidime, amoxicillin, or doripenem, as well as 1:1 combinations of the previous antibiotics with vaborbactam. Each set was repeated in triplicate, and the final viability was determined in reference to the growth control.

### 2.6. Luminescence Imaging

Luminescent images were obtained using the IVIS^®^ Lumina XRMS Series III (PerkinElmer, Waltham, MA, USA) in bioluminescence imaging mode. Exposure times were automatically determined by the LivingImage Software (version 4.5.5) with medium binning.

### 2.7. Minimum Inhibitory Concentration (MIC) Assay

MIC assays were carried out following the Clinical and Laboratory Standards Institute (CLSI) established guidelines using MHB-2 and bacterial concentrations of 5 × 10^5^ CFU/mL [57,58]. A varying concentration of quercetin derivatives or conventional antibiotics was dissolved in the media with bacteria cells. The resulting suspensions were transferred to a 96-well microliter plate at 200 μL/well (triplicate for each compound). The plate was then incubated overnight at 37 °C. The MIC was determined as the concentration at which no growth was observed.

### 2.8. In Vivo Treatment of the Murine Skin Wound Model

VIM-2 expressing bioluminescent PA (PAX5_VIM-2_) was grown overnight in LB broth. The overnight culture was then centrifuged, the media removed, and the pellet was resuspended in PBS. This suspension was later used to inoculate the mice. The experiment used fewer mice since each mouse had a paired control and test infection on the back. Three mice (JAX Swiss outbred), aged 15 weeks of age, were anesthetized with 3–5% isoflurane. Anesthesia was maintained with 2–3% isoflurane. Mice were then shaved, and the dorsal skin was scrubbed with povidone and washed away with 70% ethanol. A clear plastic template with a square 1 cm^2^ cutout was used to demarcate the corners of the planned needle-scratch grid. A 25G, 1½ needle was then used to carefully create the abrasive wound, with caution being paid not to deeply lacerate (see Appendix A for additional details). Having created the two sets of wounds, PAX5_VIM-2_ (20 µL) was added via a pipette to each wound. Mice were kept under anesthesia using the attached XGI-8 Gas Anesthesia System from Caliper LifeSciences. IVIS images were taken after 24 h, and this was determined to be the start point of the experiment (t = 0). Treatments were given while the animals were anesthetized by adding either the QATC (4:1:2, 50 µM amoxicillin, 12.5 µM clavulanate, 25 µM taxifolin in DMSO) or DMSO (control treatment) directly to the wound area at t = 0, 2, 4, 8, 24 h. IVIS images were taken once per day prior to any planned treatment. Following the final treatment at t = 24, IVIS images were taken for an additional 2 days. After that point, biopsies of the infected tissue were taken using a 6 mm biopsy punch. To each biopsy, 1 mL of PBS was added, and the tissue was homogenized. The slurry was diluted 10,000×, and 50 µL was plated onto LBA and incubated overnight at 37 °C. Colonies were counted the following day.

### 2.9. Statistical Methods

Statistical analysis was performed using OriginLab. For computational docking studies, a two-tailed unpaired *t*-test was used to test for significance between the docking of two compounds. For in vivo studies, both one-way ANOVA and paired sample *t*-tests were utilized to investigate statistical significance. All tests were performed with a *p*-value < 0.005 being considered significant. In vitro studies were performed with biological triplicates.

## 3. Results and Discussion

In order to examine the potential binding mode between quercetin analogs and VIM-2 protein, we carried out a protein–ligand docking analysis (Figure 1). The previously reported NMR-based structure of NDM-1 metallo-β-lactamase complex with quercetin (PDB: 6TTA) served as a template, demonstrating the binding modes and interactions that may be expected from the VIM-2 protein (PDB: 4C1D) since the active sites are highly conserved among these related β-lactamase enzymes (Figure 1a,b; Appendix A). The final docking model of quercetin with VIM-2 is predicted to exhibit very similar docking features as NDM-1. Some changes near the binding site of VIM-2, such as Phe62 in place of Met67 and the presence of the nearby Tyr67, further stabilize the quercetin ligand within the binding site through aromatic π-π interactions (Figure 1c). Additionally, the catechol moiety of quercetin is also able, in this static model, to hydrogen bond with the nearby Glu146. Furthermore, the Zn-O bond distances are predicted to agree with those previously determined in the NDM-1/quercetin model, suggesting stable binding to the metal ions within the active site. Together, these various factors, which appear to occur due to a more closed active site, result in several predicted poses having similar or more stable positions with acceptable glide scores compared to the NDM-1 redocked quercetin template model (Figure 1d). It is important to note that in both enzymes, the catechol moiety could be found in our model to bind to the Zn ions; however, the previously published structure of NDM-1/quercetin did not observe this binding moiety, and it has not been observed in synthetic analog complexes [27,59,60]. We have, therefore, disregarded these results, although it may be possible that they can exist for VIM-2, as the predicted docking scores were more stable with this enzyme (Appendix A).

Given the similarity of docking scores between the NDM-1 and VIM-2 bindings of quercetin, we hypothesized that quercetin and its analogs may be potentially utilized as metallo-β-lactamase inhibitors against VIM-2-expressing pathogens. Through a high-throughput screening approach, we identified 20 analogs for further in vitro testing (Figure 2). While most of these compounds are related through various amounts or locations of hydroxyl groups, several, denoted by letters b–d, were glycated.

To determine the efficacy of these compounds in vitro, a VIM-2 plasmid was transformed into a bioluminescent PA cell line (Xen 5, PerkinElmer; ATCC 19660, henceforth referred to as PAX5,) to establish PAX5_VIM-2_ cells. A monarch plasmid isolation kit (New England BioLabs, Inc.; Ipswich, MA, USA) was used to isolate a VIM-2 plasmid from in-house plasmid pet24a-VIM-2-transformed BL21(DE3) *E. coli* cells.

With this isolated plasmid, a previously established protocol for transformation via electroporation was utilized to transform the commercially acquired PAX5 to PAX5_VIM-2_ [53,61]. The successful transformation was verified utilizing both disc diffusion assays and a comparative single-dose assay against the wild type PAX5 (PAX5-WT) and PAX5_VIM-2_ (Appendix A). In both cases, the assays support the expression of the VIM-2 metallo-β-lactamase, given the fact that PAX5-WT has genes encoding for SBLs (blaOxa and blaAmpC) [62].

Based on our findings and the previous report of NDM-1 binding with quercetin (PDB ID: 6TTA), myricetin (PDB ID: 6TTAC), and morin (PDB ID: 6TT8), we hypothesized that the proper use of quercetin and its analogs would be as a part of a triple combination (QATC) containing a β-lactam (e.g., amoxicillin), an SBLi, and a quercetin analog compound. To determine the optimal ratio of QATC, we first determined the MIC of amoxicillin to be 100 µM or 36 µg/mL, followed by the MIC of clavulanate with amoxicillin (Appendix A).

Observing that a 1:1 molar ratio of these compounds was enough to inhibit PAX5_VIM-2_ cell growth, we chose to test the efficacy of our chosen analogs in combination with a 2:1 molar ratio of amoxicillin to clavulanate (a roughly 7:1 ratio by mass, which is intermediate in the possible ratios of augmentin; Figure 3a). The single-dose results indicated that many of the analogs performed similarly well, with the majority of those tested nearly doubling the efficacy of the QATC vs. augmentin. Quercetin analogs with more than 90% of cell growth inhibition were chosen for further testing in order to determine the MIC of their respective QATCs (Figure 3b). MIC analysis revealed that compounds **#4** (quercetagetin), **#5** (taxifolin), and **#14** (chrysoeriol) were the most potent inhibitors against PAX5_VIM-2_. Importantly, glycation, although predicted to be a potentially useful strategy for inhibition, was found to be generally less effective. Furthermore, we determined that many of these investigational compounds were effectively non-toxic when tested against wild type human HEK 293T cells (Figure 3b, Appendix A).

To better determine its enzyme-level kinetic inhibition activity and target specificity via its binding mode, the kinetic assays of VIM-2 inhibition and computational docking studies were performed for these compounds for comparison (Figure 4).

Kinetics studies were carried out using nitrocefin as a colorimetric indicator, with the hydrolyzed product measured at 486 nm [52]. The K_M_ of our VIM-2 was found to be higher than what is typically reported; however, this may be a result of our use of PBS rather than other buffers, such as HEPES [63]. The discrepancy is also unlikely to be related to metalation, as low Zn, one Zn, and two Zn kinetics have been previously determined, and the K_M_ values have been observed to vary only slightly [6]. Instead, it was typically observed that *k_cat_* was more significantly affected; however, two Zn *k_cat_* values agree with our determined result of 18.85 ± 1.96 s^−1^ (Appendix A).

In agreement with the previously shown MIC values, **#4**, **#5,** and **#13** were found to be the best inhibitors, with no statistical difference in their K_i_ values (Figure 4a). However, **#5**, which consistently showed reproducible inhibition, was selected as a candidate for further investigation. In the docking study, **#5** was found to bind in a similar fashion as **#1a** (quercetin) at the binding site of the VIM-2 enzyme compared with the previous modeling study (Figure 4b). The predicted docking poses indicate that the nearby Asp117 and Arg205 participate in hydrogen binding, helping to stabilize the inhibitor within the active site (Appendix A). These factors appear to result in a significantly lower (better) docking score for **#5** than **#1a** (−8.01 ± 0.57 kcal/mol vs.−6.74 ± 0.48 kcal/mol, *p* < 0.01, Figure 4c). It is also likely that the greater flexibility of **#5**, owing to its reduced unsaturation compared to **#1a,** allows this molecule to adopt more conformations that are more favorable to fit into the binding site of the enzyme. Additionally, the ligand metal bond distances are predicted to be similar to those found in the published 6TTA crystal structure, with an average bond length of 2.70 Å vs. 2.64 Å, respectively (Figure 4d). The slightly longer average bond distance for the predicted **#5**-VIM-2 model may be explained by the more neutral position of **#5** between the two Zn ions, whereas quercetin in the 6TTA structure is much closer and angled toward Zn(1). Recently, a report was published describing the potential of quercetin and its analog compounds to act as SBLis [64]. Throughout our investigation, we were unable to observe significant inhibition without the addition of clavulanate, suggesting that these compounds may only have very narrow SBL coverage. Having only purified TEM-1 available (class A), we tested the single-dose inhibition of this enzyme by these analogs in comparison with established SBLis (clavulanate, sulbactam, vaborbactam, and tazobactam; Appendix A). No inhibition by analog compounds was observed, suggesting that these compounds, if able to inhibit some SBLs, may have limited coverage and may still be better suited as MBLis. Furthermore, we confirmed with SBL-only PAX5-WT that taxifolin was unable to provide any significant benefit in the killing. This further supports our hypothesis that these polyphenols are predominantly MBLis (Appendix A).

To explore the therapeutic potential of **#5** as a VIM-2 inhibitor, we used a murine skin wound infection model to perform a test of QATC. This model was chosen for several reasons. (i) The use of a localized, small infection allows for a significant reduction in the number of mice as well as a direct comparison between treatments and controls, (ii) The direct application of a test compound to the skin allows for more rapid efficacy testing without requiring significant efforts to be spent on potential solubility, absorption, metabolism, or excretion problems. (iii) Skin and soft tissue infections with PA are common with >10% isolation while typically resulting in greater lengths of stay and patient cost due to frequently inappropriate treatment [65,66,67]. Furthermore, without proper therapy, skin and soft tissue infections (SSTI) can progress and become life-threatening. Additionally, multiple strategies exist on how to generate skin wounds. These differ by the depth and regularity of the wound and may consist of burns, abrasions, superficial or deep incisional wounds, and excisional wounds extending to the underlying muscle and fascia [66,68,69]. Previous reports have utilized these various methodologies with PA at concentrations ranging from 10^2^ to 10^8^ CFU. Based on these publications, we chose a needle-scratch method for generating an abrasive skin wound, as this approach has been reported to result in more stable infections with PA compared to an easier-to-perform excisional approach.

Briefly, two square grids of needle scratches (each having an area of 1 cm^2^) were created on the backs of each mouse. Each grid was inoculated with PAX5_VIM-2_ (20 μL, 2.4 × 10^9^ CFU, in PBS; Figure 5a,b, Appendix A; Appendix A). IVIS images were taken once daily prior to any treatments. Based on our previous in vitro studies, the QATC combination was prepared to contain 50 µM amoxicillin, 12.5 µM clavulanate, and 25 µM taxifolin. DMSO was used as the control. Treatments were applied daily for four days. Bioluminescence images were taken daily, and the radiative flux (photons/sec/cm^2^) was tracked as a measure of how the bacterial concentration changed over time (Appendix A). In both cases, the treatment and control populations decreased with time, but the wounds treated with the QATC decreased to a greater extent due to the initially higher bacterial concentration in the infection (Figure 5b).

The mice skin wound infectious model test was performed in triplicate, and the corresponding IVIS mice image was successfully acquired (Appendix A). As evidenced by the large difference in bioluminescent signal intensity, we consider the QATC therapy to be promising for treating *P. aeruginosa,* which maintains sustained expression of VIM-2 β-lactamases. QATC-treated infections showed significant variation in ROI reduction, indicating that most bacteria were effectively killed, and QATC treatment appears to reduce bacterial load (based on bioluminescent intensity) compared with the control.

## 4. Conclusions

Through combined in silico modeling, in vitro, and in vivo approaches, we have demonstrated that quercetin analog compounds, specifically taxifolin, are promising candidates for further derivitization and development into VIM-2 MBLis. In addition to the measured low toxicity, the utilization of taxifolin as a VIM-2 MBLi is a promising lead for expanding the use of current FDA-approved SBL-targeting combination treatments. The current lack of approved MBLis requires continued innovation and discovery of alternative structures that can inhibit the action of MBLs and be combined with current SBLs into triple therapies that are capable of addressing the threat of drug-resistant ESKAPE pathogens while also preserving the use of β-lactam antibiotics.

## Figures and Tables

**Figure 1 microorganisms-11-02653-f001:**
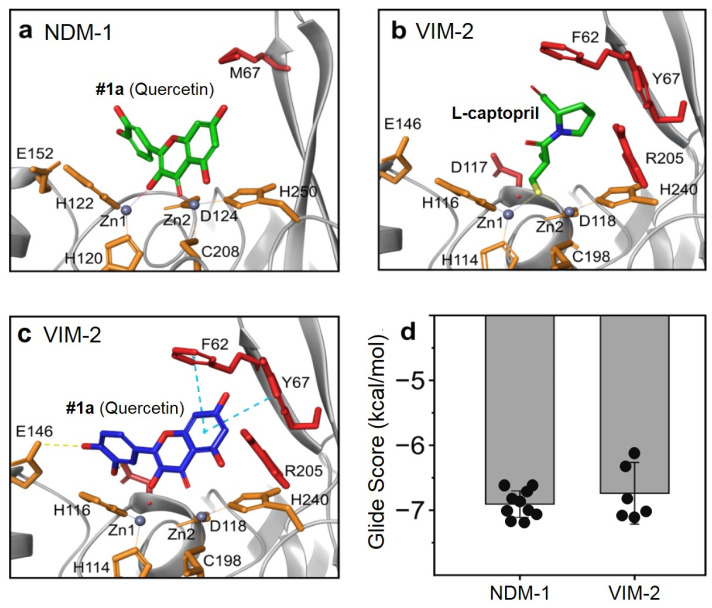
Docking study of quercetin with both NDM-1 and VIM-2. (**a**) NMR structure of quercetin at the active site of NDM-1 (PDB:6TTA). (**b**) X-ray crystal structure of the metallo-β-lactamase VIM-2 with L-captopril (PDB: 4C1D). (**c**) Predicted binding mode of quercetin with VIM-2 active site. (**d**) Comparison of glide docking scores for quercetin with both NDM-1 and VIM-2. Orange residues have similar locations in both NDM-1 and VIM-2; red residues are different.

**Figure 2 microorganisms-11-02653-f002:**
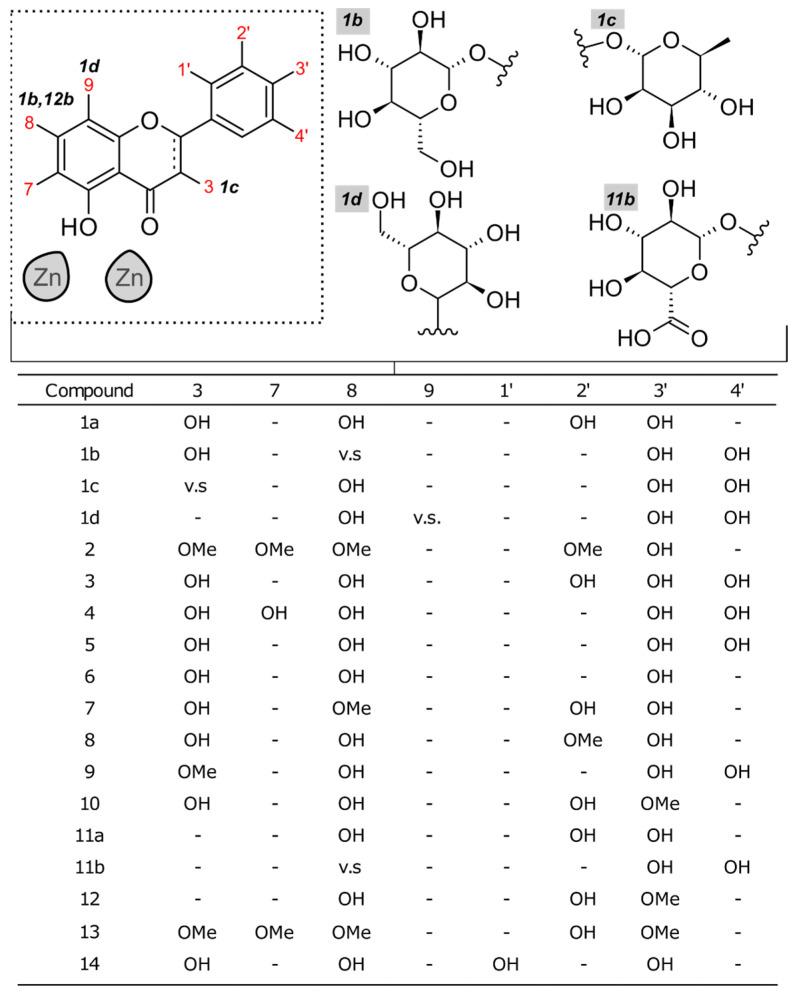
Structure variety of the quercetin analogs investigated. Cells labeled with v.s. (vide supra) indicate that modification is drawn above.

**Figure 3 microorganisms-11-02653-f003:**
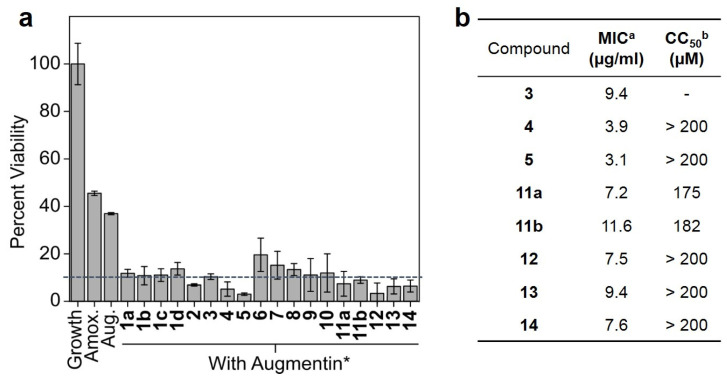
Cell-based studies against PAX5_VIM-2_. (**a**) Single-dose inhibition assays were performed for 18 h. Amoxicillin and augmentin were used as controls, and the compounds were treated in combination with augmentin. All tests were triplicated. (**b**) Toxicity values of Augmentin+quercetin analogues and of quercetin analogues alone against human kidney cell lines. ^a^ quercetin analogs in combination with augmentin. ^b^ quercetin analogs alone. * Augmentin was 25 µM amoxicillin and 12.5 µM clavulanate (2:1 ratio) and was tested against HEK293Twt cells.

**Figure 4 microorganisms-11-02653-f004:**
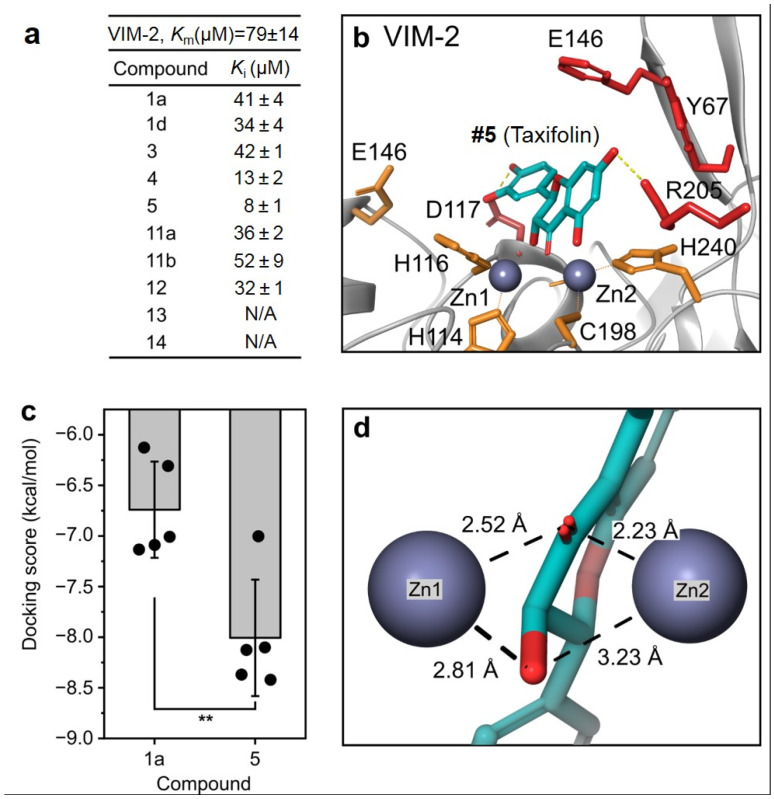
Enzyme kinetics and the protein–ligand docking study. (**a**) Measured K_M_ and K_i_ values of quercetin analogs against VIM-2 with its colorimetric substrate, nitrocefin. (**b**) Predicted binding mode of compound **#5** with the VIM-2 (PDB: 4C1D) enzyme. (**c**) Docking score comparison between compound **#1a** (quercetin) and **#5** (taxifolin). (**d**) Predicted bond distance between compound **#5** and Zn(1) and Zn(2) at the active site of the VIM-2 enzyme. ** indicates *p* < 0.01.

**Figure 5 microorganisms-11-02653-f005:**
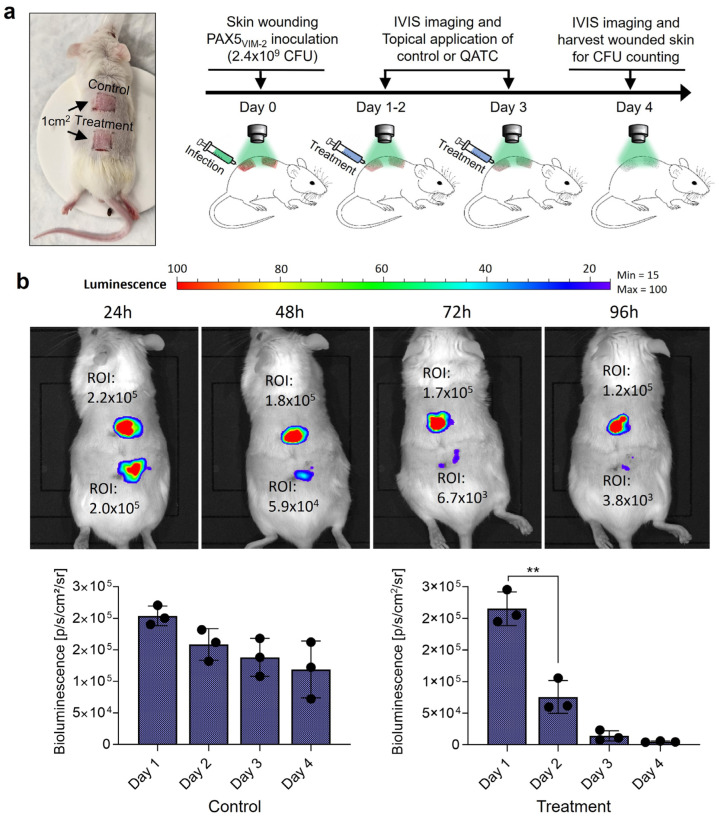
*In vivo* study of murine wound models infected with PAX5_VIM-2_. (**a**) The treatment scheme of wound infections induced by PAX5_VIM-2_ in a murine model following 4 days of treatment for the vehicle control or augmentin+**#5**. (**b**) IVIS imaging of local skin wound infection by bioluminescent *P. aeruginosa* (Xen05, VIM-2(+)) treatment. Four days of infection status comparison was quantified via bioluminescence ROI for each treatment. The bottom panel bar graphs show the quantification of the average bioluminescence intensity emitted by bacteria. Data are presented as mean ± SD (n = three independent mice, ** *p* < 0.005, ANOVA test).

## Data Availability

The datasets generated during and/or analyzed during the current study are available from the corresponding author upon reasonable request.

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
