# Peer review of "Taxifolin as a Metallo-β-Lactamase Inhibitor in Combination with Augmentin against Verona Imipenemase 2 Expressing Pseudomonas aeruginosa"

_microorganisms, 2023, doi:10.3390/microorganisms11112653_

Round 1
Reviewer 1 Report
Comments and Suggestions for Authors
The presence of Serine- β-lactamase (SBL) and metallo-β-lactamase (MBL) are major causes for the resistance of β-lactam antibiotics. As a counter approach, SBL inhibitor clavulanate is combined with amoxicillin to promote antibiotic action by protecting the SBL-mediated hydrolysis of the β-lactam ring.
However, a suitable MBL inhibitor is absent at the clinical level. The current work has shown a glimpse of MBL inhibitors in the form of flavonol compounds, such as taxifolin.
Authors have demonstrated taxifolin potentiates the antibiotic action of amoxicillin and clavulanate combination by promoting its bacterial killing in a VIM2-MBL expressing P. aeruginosa recombinant strain. Enzymatic activity assay also points to MBL as a possible target for taxifolin.
However, it is not clear that the mechanistic link is limited to MBL only. Some recent reports described that quercetin and its analog compounds act as SBL. The current experimental approach (Fig.3a) did not rule out a taxifolin-SBL link. In order to do that, the authors must use a strain that does not express MBL, but express SBL (e.g, PAX5-WT or any SBL overexpressing stain) and address the following questions:
Did augmentin promote amoxicillin killing in strains lacking MBL, but expressing SBL (PAX5-WT)?
Does taxifolin promote amoxicillin killing in strains lacking MBL, but expressing SBL (PAX5-WT)?
1. Sensitivity (Percent killing) of PAX5-WT to triple combination.
What accounts for the pivotal role of clavulanate in the effectivity of the triple combination against MBL overexpressing strain, remains unclear. Even the speculated SBL expression (inherited from WT) in the recombinant strain may not fully explain it.
Perhaps, it is worth to test the sensitivity of triple combination in MBL lacking, but SBL expressing strain to rule out taxifolin influence on clavulanate.
Comments on the Quality of English LanguageNo problem with the quality of English language.
Author Response
Dear Ms. Liang,
We have revised the manuscript considering the Reviewers’ comments. We feel that we were able to satisfactorily address the comments of both Reviewers and have worked to improve the clarity of our writing as suggested by the Reviewers.
Specifically, we have performed an additional single-dose cell viability assay to support the idea that taxifolin only targets MBLs rather than SBLs. This figure has been added to the Supplementary Information and has been referred to within the Main Text. Additionally, we have modified the writing in the manuscript, taking into account many of the corrections given by Reviewer 2.
Below, we detail the revisions made to the manuscript and our responses to the comments of all Reviewers. While hoping for positive feedback, we will be grateful for further comments and suggestions.
Sincerely,
Woo Shik Shin, on behalf of all authors.
Answers are in blue; actions are in green. Replies are numbered (e.g. 1.x and 2.x for the two Reviewers). To see the changes made to the manuscript, please see the comments and highlighted text in the revised manuscript. Those comments link to the numbered replies below.
Reviewer #1 (Remarks to the Author):
The presence of Serine- β-lactamase (SBL) and metallo-β-lactamase (MBL) are major causes for the resistance of β-lactam antibiotics. As a counter approach, SBL inhibitor clavulanate is combined with amoxicillin to promote antibiotic action by protecting the SBL-mediated hydrolysis of the β-lactam ring.
However, a suitable MBL inhibitor is absent at the clinical level. The current work has shown a glimpse of MBL inhibitors in the form of flavonol compounds, such as taxifolin.
Authors have demonstrated taxifolin potentiates the antibiotic action of amoxicillin and clavulanate combination by promoting its bacterial killing in a VIM2-MBL expressing P. aeruginosa recombinant strain. Enzymatic activity assay also points to MBL as a possible target for taxifolin.
Reply: We would like to thank the Reviewer for their time and their positive and constructive feedback.
Comment 1.1: However, it is not clear that the mechanistic link is limited to MBL only. Some recent reports described that quercetin and its analog compounds act as SBL. The current experimental approach (Fig.3a) did not rule out a taxifolin-SBL link. In order to do that, the authors must use a strain that does not express MBL, but express SBL (e.g, PAX5-WT or any SBL overexpressing stain) and address the following questions: Did augmentin promote amoxicillin killing in strains lacking MBL, but expressing SBL (PAX5-WT)? Does taxifolin promote amoxicillin killing in strains lacking MBL, but expressing SBL (PAX5-WT)? What accounts for the pivotal role of clavulanate in the effectivity of the triple combination against MBL overexpressing strain, remains unclear. Even the speculated SBL expression (inherited from WT) in the recombinant strain may not fully explain it.
Perhaps, it is worth to test the sensitivity of triple combination in MBL lacking, but SBL expressing strain to rule out taxifolin influence on clavulanate.
Reply 1.1: Indeed, the reviewer brings up a good point that the potential inhibition of the SBLs produced by PAX5 (AmpC, OXA) were not completely ruled out. Although we have previously demonstrated that our polyphenol inhibitors were unable to prevent TEM from hydrolyzing nitrocefin (Supp. Fig. S8).
These results clearly demonstrate that Class A SBLs are impervious to inhibition by these compounds.
Action 1.1: To further clarify whether these are suitable inhibitors for SBLs produced by our PA species, we performed an additional single-dose experiment in which the treatments mirrored those utilized throughout the manuscript: amoxicillin (25 μM), amoxicillin+clavulanate (25 μM/12.5 μM), amoxicillin+taxifolin (25 μM/25 μM), and QATC (25 μM/12.5 μM/25 μM).
We have also added the following sentence to the main text:
“We furthermore confirmed with SBL-only PAX5-WT that taxifolin was unable to provide any significant benefit in killing. This further supports our hypothesis that these polyphenols are predominantly MBLis (Supplementary Figure S9).”
In closing, we again thank the Reviewers for their comments and suggestions. We hope that we have addressed their concerns, and we will be happy to answer any further questions as needed. We also thank the Editor for considering our revised manuscript.
Sincerely,
Woo Shik Shin and all co-authors.
Reviewer 2 Report
Comments and Suggestions for Authors
Use Acrobat Reader to see my comments.

Moderate English language editing required
Author Response
Dear Ms. Liang,
We have revised the manuscript considering the Reviewers’ comments. We feel that we were able to satisfactorily address the comments of both Reviewers and have worked to improve the clarity of our writing as suggested by the Reviewers.
Specifically, we have performed an additional single-dose cell viability assay to support the idea that taxifolin only targets MBLs rather than SBLs. This figure has been added to the Supplementary Information and has been referred to within the Main Text. Additionally, we have modified the writing in the manuscript, taking into account many of the corrections given by Reviewer 2.
Below, we detail the revisions made to the manuscript and our responses to the comments of all Reviewers. While hoping for positive feedback, we will be grateful for further comments and suggestions.
Sincerely,
Woo Shik Shin, on behalf of all authors.
Answers are in blue; actions are in green. Replies are numbered (e.g. 1.x and 2.x for the two Reviewers). To see the changes made to the manuscript, please see the comments and highlighted text in the revised manuscript. Those comments link to the numbered replies below.
Reviewer #2 (Remarks to the Author): We thank the Reviewer for the detailed review of our manuscript and for their suggested corrections.
Comment 2.1: Materials and methods section, Luminescence imaging subsection, Lines 182-185: describe more in detail the in vivo experiment since some fundamental information are missing ( e.g. numerosity, age, strain of mice). Control group how was designed? how many animals are tested? have you worked with triplicates? could you provide the ethical statement?
Reply 2.1: This section is not meant to describe the animal experiment, but rather just the luminescence imaging. This is because luminescence imaging was also used to image vials containing the luminescent bacteria. The details of the in vivo experiment are described later in the materials and methods section.
Action 2.1: To increase clarity, we have moved the sentence regarding mice and anesthesia to the in vivo mouse study section.
Comment 2.2: Materials and methods section, in vivo treatment of murine skin wound model subsection, Line 195: what's PAX5VIM-2? add a brief description
Reply 2.2: In this sentence, this abbreviation is written as " VIM-2 expressing bioluminescent PA". The PA abbreviation is also previously written in the introduction.
Action 2.2: We have added a sentence to the transformation paragraph to help explain this abbreviation: "Starting with bioluminescent PA (PAX5), we developed a VIM-2 expressing strain of PAX5 (PAX5VIM-2)."
Comment 2.3: Materials and methods section, in vivo treatment of murine skin wound model subsection, Line 197: which is the strain used? Don't you think that three animals are quite low to draw back any statistics?
Reply 2.3: Generally, three animals would be very little. However, we have utilized a paired model in which each animal was a test and control subject. This was possible since this is a localized skin-wound infection model. Since the scratches are superficial (Supplementary note 1), we were able to apply treatments and the vehicle control to the same mouse and therefore better control for potential differences between mice.
Comment 2.4: Materials and methods section, in vivo treatment of murine skin wound model subsection, Line 207: Why have you used DMSO?
Reply 2.4: DMSO was the solvent in which all antibiotics and test compounds were dissolved. It was therefore used as a vehicle control.
Comment 2.5: Results and discussions section, Lines 264-275: this paragraph should be moved in M&M section since it reports the entire protocol of transformation (and report the PAX strain cited in "in vivo" paragraph without previous presentation)
Reply 2.5: These details have already been written and mentioned in the materials and methods section. This section is not a reproduction of the protocol but rather a reminder as to how this strain was obtained since it is not commercially available. We prefer to keep such details so that the interested reader need not scroll between this section and the method and so that there is less confusion regarding which strain is utilized and how it was obtained.
Comment 2.6: Results section, Lines 329-338: mantain coherence when cite your compounds (prefer the entire name insted of number)
Action 2.6: To increase coherence, we have replaced the varied naming with simply "#5" after the first explanation of this notation where it is written that this is taxifolin.
Comment 2.7: Results and Discussion section, Lines 374-382: move this section to M&M, here is useless
Reply 2.7: These details have already been written and mentioned in the materials and methods section. This section is not a reproduction of the protocol but rather a reminder as to the important parameters of the experiment. We prefer to keep such details so that the interested reader need not scroll between this section and the method.
Comment 2.8: Figure 5: split this panle. Panel a should be moved to M&M section since is perfectly summarize the experimental protocol used. Panel b reports your results so that it can be maintained here
Reply 2.8: We agree that the top panel summarizes our method. For this reason, we have chosen to keep everything together. For readers who are not experts in this field, the side-by-side demonstration of the method and the results will allow for greater comprehension.
In closing, we again thank the Reviewers for their comments and suggestions. We hope that we have addressed their concerns, and we will be happy to answer any further questions as needed. We also thank the Editor for considering our revised manuscript.
Sincerely,
Woo Shik Shin and all co-authors.

Round 2
Reviewer 2 Report
Comments and Suggestions for Authors
Thank you for the revised version provided. I have to point out that the ethicalt statement for the in vivo experiment is missing.
Author Response
Dear Ms. Liang and the reviewer,
Based on the reviewers' comments, a missing "ethical statement" regarding the in vivo experiments was added to the manuscript. While hoping for positive feedback, we will be grateful for further suggestions.
Ethical statements
All animal procedures were performed in accordance with the Guide for Care and Use of Laboratory Animals and approved by the Institutional Animal Care & Use Committee (IACUC, Protocol # 21-03-295) of the Department of Pharmaceutical Sciences, College of Pharmacy, Northeast Ohio Medical University.
Respectfully,
Woo Shik Shin, on behalf of all authors.
